# SOFT: Softmax-free Transformer with Linear Complexity

**Jiachen Lu**[1]    **Jinghan Yao**[1]    **Junge Zhang**[1]    **Xiatian Zhu**[2]    **Hang Xu**[3]
**Weiguo Gao**[1]    **Chunjing Xu**[3]    **Tao Xiang**[2]    **Li Zhang**[1]*

[1]Fudan University    [2]University of Surrey    [3]Huawei Noah's Ark Lab

https://fudan-zvg.github.io/SOFT

## Abstract

Vision transformers (ViTs) have pushed the state-of-the-art for various visual recognition tasks by patch-wise image tokenization followed by self-attention. However, the employment of self-attention modules results in a quadratic complexity in both computation and memory usage. Various attempts on approximating the self-attention computation with linear complexity have been made in Natural Language Processing. However, an in-depth analysis in this work shows that they are either theoretically flawed or empirically ineffective for visual recognition. We further identify that their limitations are rooted in keeping the *softmax self-attention* during approximations. Specifically, conventional self-attention is computed by normalizing the scaled dot-product between token feature vectors. Keeping this softmax operation challenges any subsequent linearization efforts. Based on this insight, for the first time, a *softmax-free transformer* or SOFT is proposed. To remove softmax in self-attention, Gaussian kernel function is used to replace the dot-product similarity without further normalization. This enables a full self-attention matrix to be approximated via a low-rank matrix decomposition. The robustness of the approximation is achieved by calculating its Moore-Penrose inverse using a Newton-Raphson method. Extensive experiments on ImageNet show that our SOFT significantly improves the computational efficiency of existing ViT variants. Crucially, with a linear complexity, much longer token sequences are permitted in SOFT, resulting in superior trade-off between accuracy and complexity.

## 1   Introduction

Recently the step change brought by Transformers [33] in natural language processing (NLP) [10, 4] seems to have arrived in vision [11, 41, 47, 46]. Indeed, with less inductive bias in its architecture design than Convolution neural networks (CNNs), pure Vision Transformer (ViT) [11] and its variants have shown to be able to outperform CNNs on various vision tasks [8, 15]. However, there is a bottleneck in any Transformer based model, namely its quadratic complexity in both computation and memory usage. This is intrinsic to the self-attention mechanism: given a sequence of tokens (*e.g.*, words or image patches) as input, the self-attention module iteratively learns the feature representations by relating one token to all other tokens. This results in a quadratic complexity $O(n^2)$ with the token sequence length $n$ in both computation (time) and memory (space) since an $n \times n$ sized attention matrix needs to be computed and saved during inference. This problem is particularly acute in vision: a 2D image after tokenization will produce a far longer sequence than those in NLP even with a moderate spatial resolution. This quadratic complexity thus prevents a ViT model from modeling images at high spatial resolutions, which are often crucial for visual recognition tasks.

---

*Li Zhang (lizhangfd@fudan.edu.cn) is the corresponding author with School of Data Science, Fudan University.

35th Conference on Neural Information Processing Systems (NeurIPS 2021).

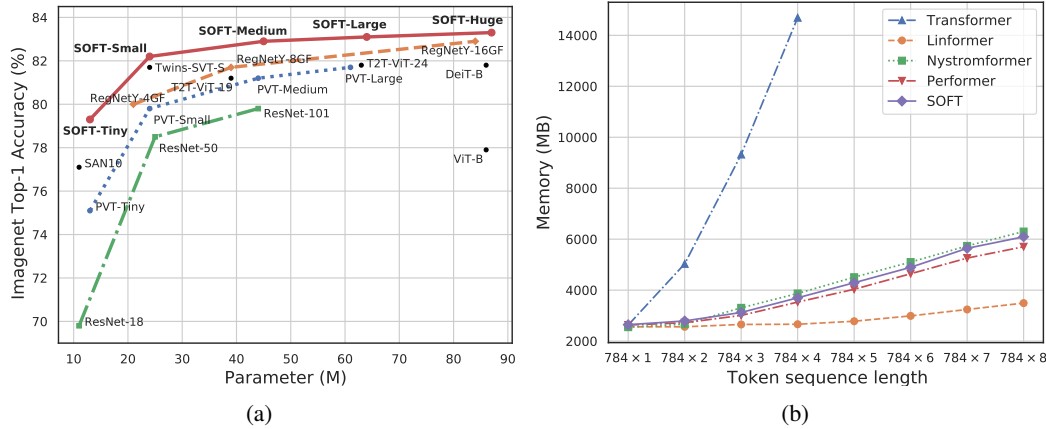

Figure 1: Top1-Accuracy on ImageNet [9] validation set with respect to parameters and the memory usage corresponding to the token sequence length in practice compared to other methods. (a) Comparison with CNN models: RegNet [26], ResNet [13] and Transformer models: PVT [35], DeiT [31], ViT [11], T2T-ViT [41], Twins-SVT [6] and SAN10 [45]; (b) Comparison with Transformer [33], Linformer [34], Nyströformer [39] and Performer [5]. The memory usage is measured with a batch size of 1 on a 16GB Tesla V100.

A natural solution is to reduce the complexity of self-attention computation via approximation. Indeed, there have been a number of attempts in NLP [34, 5, 18, 39]. For example, [34] takes a naive approach by shortening the length of Key and Value via learnable projections. Such a coarse approximation would inevitably cause performance degradation. In contrast, [5, 17] both leverage the kernel mechanism to approximate softmax normalization to linearize the computation in self-attention. [18] instead adopts a hashing strategy to selectively compute the most similar pairs. Recently, [39] uses Nyström matrix decomposition to reconstruct the full attention matrix with polynomial iteration for approximating the pseudo-inverse of the landmark matrix. Nonetheless, softmax normalization is simply duplicated across the matrix decomposition process, which is theoretically unsound. We empirically found that none of these methods are effective when applied to vision (see Sec. 4.2).

In this work, we identify that the limitations of existing efficient Transformers are caused by the use of *softmax self-attention*, and for the first time propose a softmax-free Transformer. More specifically, in all existing Transformers (with or without linearization), a softmax normalization is needed on top of scaled dot-product between token feature vectors [33]. Keeping this softmax operation challenges any subsequent linearization efforts. To overcome this obstacle, we introduce a novel *softmax-free self-attention* mechanism, named as SOFT, with linear complexity $O(n)$ in both space and time. Specifically, SOFT uses Gaussian kernel to define the similarity (self-attention) function without the need for subsequent softmax normalization. With this softmax-free attention matrix, we further introduce a novel low-rank matrix decomposition algorithm for approximation. The robustness of the approximation is theoretically guaranteed by employing a Newton-Raphson method for reliably computing the Moore-Penrose inverse of the matrix.

We make the following **contributions**. **(I)** We introduce a novel *softmax-free Transformer* with linear space and time complexity. **(II)** Our attention matrix approximation is achieved through a novel matrix decomposition algorithm with theoretical guarantee. **(III)** To evaluate our method for visual recognition tasks, we design a family of generic backbone architectures with varying capacities using SOFT as the core self-attention component. Extensive experiments show that with a linear complexity (Figure 1b), our SOFT models can take in as input much longer image token sequences. As a result, with the same model size, our SOFT outperforms the state-of-the-art CNNs and ViT variants on ImageNet [9] classification in the accuracy/complexity trade-off (Figure 1a).

## 2 Related work

**Vision Transformers** There is a surge of research interests recently in exploiting Transformers for visual recognition tasks [36, 35, 41, 31, 44], inspired by their remarkable success in NLP [33, 10, 4].

Core to these NLP and vision transformers is the same self-attention mechanism [33] that computes a self-attention matrix by exhaustively comparing token pairs. This means a quadratic complexity with the sequence length in both space and time, which thus limits the scalability of Transformers in dealing with long sequences. This limitation is more serious in vision than NLP: To process an image with at least thousands of pixels, patch-wise tokenization is a must for Transformers to control the computational cost. Given higher resolution images, the patch size also needs to be enlarged proportionally sacrificing the spatial resolution. This limits the capability of Transformers, *e.g.*, learning fine-grained feature representation as required in many visual recognition tasks.

**Linear Transformers** Recently, there have been a number of linear/efficient variants [5, 34, 17, 18, 30, 24, 16] of Transformers in NLP. For example, [34] learns to shrink the length of Key and Value based on a low-rank assumption. [18] adopts a hashing strategy to selective the most similar pairs and only compute attention among them. [5, 17] utilize different kernel functions for approximating softmax-based self-attention matrix. [24] applies random feature mapping on the sequences to approach the original softmax function. [16] decreases the time and memory consumption of the attention matrix by replacing the softmax function with its linear-complexity recurrent alternative. When applied to visual recognition tasks, however, we show that these models have considerable performance degradation compared to the standard Transformers [33] (see Sec. 4.2).

The most related work to SOFT is [39] which uses the Nyström matrix decomposition to avoid computing the full attention matrix. However, this method suffers from several theoretical defects: (1) As the standard self-attention needs to apply row-wise softmax normalization on the full attention matrix, a direct application of matrix decomposition is infeasible. As a workaround, softmax is simply applied to all the ingredient matrices in [39]. Such an approximation is not guaranteed theoretically. (2) With a polynomial iteration method, it is not guaranteed that the generalized attention matrix inverse can be computed when the matrix is a nearly singular one in practice. In contrast to all the above methods, in this paper we propose a *softmax-free* self-attention mechanism that facilitates matrix decomposition for complexity minimization with theoretical guarantees.

# 3 Method

## 3.1 Softmax-free self-attention formulation

A schematic illustration of our model is given in Figure 2. Let's first look at our attention module design. Given a sequence of $n$ tokens $X \in \mathbb{R}^{n \times d}$ with each token represented by a $d$-dimensional feature vector, self-attention [33] aims to discover the correlations of all token pairs exhaustively.

Formally, $X$ is first linearly projected into three $d_e$-dimensional spaces (query, key, and values) as:

$$Q = XW_q \in \mathbb{R}^{n \times d_e}, \quad K = XW_k \in \mathbb{R}^{n \times d_e}, \quad V = XW_v \in \mathbb{R}^{n \times d_e}, \tag{1}$$

where $W_q, W_k, W_v \in \mathbb{R}^{d \times d_e}$ are learnable matrices. Self-attention can be expressed in a generic formulation as:

$$y_{i,:} = \sum_{j=1}^{n} \alpha(Q_{i,:}, K_{j,:}) \odot V_{j,:}, \tag{2}$$

where $\odot$ is the Hadamard product, and $i, j \in \{1, \cdots, n\}$ index the tokens. The key self-attention function $\alpha : \mathbb{R}^{d_e} \times \mathbb{R}^{d_e} \to \mathbb{R}$ is composed of a nonlinear function $\beta : \mathbb{R} \to \mathbb{R}$ and a relation function $\gamma : \mathbb{R}^{d_e} \times \mathbb{R}^{d_e} \to \mathbb{R}$. A dominant instantiation of $\alpha$ is the scaled dot-product based softmax self-attention [33], defined as

$$\beta(\cdot) = \text{softmax}(\cdot), \quad \gamma(Q_{i,:}, K_{j,:}) = \frac{1}{\sqrt{d_e}} \cdot Q_{i,:}^\top K_{j,:}. \tag{3}$$

Whilst this softmax self-attention has been the *de facto* choice and seldomly questioned, as discussed earlier it is not necessarily suited for linearization. To facilitate the design of linear self-attention, we introduce a softmax-free self-attention function with the dot-product replaced by a Gaussian kernel as:

$$\beta'(\cdot) = \exp(\cdot), \quad \gamma'(Q_{i,:}, K_{j,:}) = -\frac{1}{2\sqrt{d_e}} \cdot \|Q_{i,:} - K_{j,:}\|_2^2. \tag{4}$$

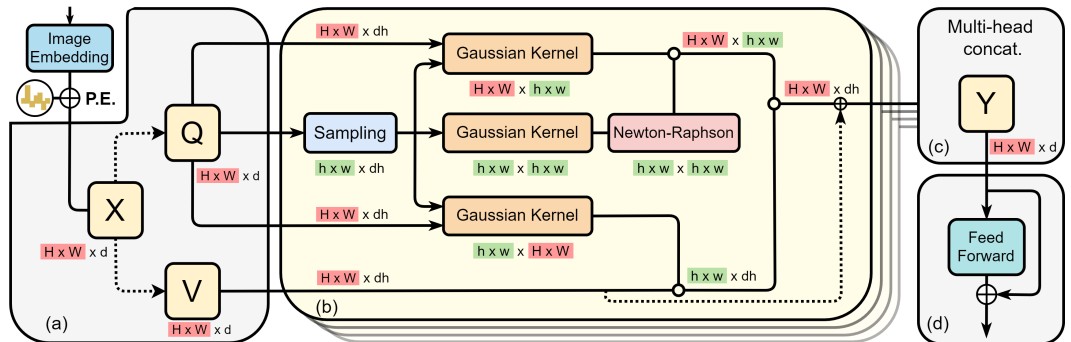

Figure 2: Schematic illustration of the proposed softmax-free self-attention (SOFT) method. `P.E.`: Position embedding. Dash lines: linear projection. `dh`: the hidden dim of each attention head. ∘ denotes the matrix dot product.

To preserve the symmetric property of attention matrix as in Eq (3), we set the project matrices $W_q$ and $W_k$ in Eq (1) identical (*i.e.*, $Q = K$). Our self-attention matrix is then written as:

$$S_{i,j} = \exp\left(-\frac{1}{2\sqrt{d_e}} \cdot \|Q_{i,:} - K_{j,:}\|_2^2\right). \tag{5}$$

For notation simplicity, we define the matrix formulation as: $S = \exp(Q \ominus K)$.

**Remarks** Our self-attention matrix $S$ has three important properties: (1) It is symmetric; (2) All the elements lie in a unit range of $[0, 1]$; (3) All diagonal elements hold the largest value 1 (self-reinforced), with the bottom ones (corresponding to most dissimilar token pairs) being close to 0. As Gaussian kernel is a positive definite kernel [12], $S$ is deemed a Gram matrix. However, we find that when using our kernel-based self-attention matrix $S$ without linearization, the training of a transformer fails to converge. This might explain why softmax dot-product based self-attention [33] is so popular in vanilla transformers.

### 3.2 Low-rank regularization via matrix decomposition with linear complexity

To solve the convergence and quadratic complexity problems, we leverage matrix decomposition as a unified solution with low-rank regularization. In particular, we consider Nyström [38], which is originally a low-rank matrix approximation algorithm. This enables our model's complexity to be reduced significantly without computing the full self-attention matrix $S$.

We make this choice because our $S$ is positive semi-definite (*i.e.*, a Gram matrix) without follow-up normalization which are all necessary conditions for Nyström. In contrast, [39] totally ignores these requirements, leading to theoretical flaw in its approximation.

To define the Nyström method formally, let us express $S = \exp(Q \ominus K)$ as a block matrix:

$$S = \begin{bmatrix} A & B \\ B^\top & C \end{bmatrix} \in \mathbb{R}^{n \times n}, \tag{6}$$

where $A \in \mathbb{R}^{m \times m}$, $B \in \mathbb{R}^{m \times (n-m)}$, $C \in \mathbb{R}^{(n-m) \times (n-m)}$ with $m \ll n$. Through Nyström decomposition (see derivative details in Supplementary A.1), an approximation can be represented as:

$$\hat{S} = \begin{bmatrix} A \\ B^\top \end{bmatrix} A^\dagger \begin{bmatrix} A & B \end{bmatrix} = P^\top A^\dagger P, \quad \text{where} \quad P = \begin{bmatrix} A & B \end{bmatrix}, \tag{7}$$

and $A^\dagger$ is the Moore-Penrose (a generalized) inverse of $A$.

**Sampling** In the standard Nyström formulation, $A$ and $B$ are sub-matrices of $S$ obtained by randomly sampled $m$ tokens, denoted as $\widetilde{Q}$. We call the sampled $\widetilde{Q}$ as bottleneck tokens. However,

| **Algorithm 1:** SOFT: Softmax-free attention | **Algorithm 2:** NR: Newton-Raphson iteration |
|---|---|
| **Input:** $Q \in \mathbb{R}^{n \times d_e}$, sampling function $f_s$ | **Input:** $A \in \mathbb{R}^{m \times m}$, and $\mathcal{T} \in \mathbb{Z}^+$ |
| **Sampling** $\widetilde{Q} \leftarrow f_s(Q)$ ; | $\alpha = 2/\|A\|_1^2$.Initialize $A_0 \leftarrow \alpha A$; |
| $A \leftarrow \exp(\widetilde{Q} \ominus \widetilde{Q}), P \leftarrow \exp(\widetilde{Q} \ominus Q)$; | **for** $k$ *from* $1$ *to* $\mathcal{T}$ **do** |
| $\hat{S} \leftarrow P^\top \text{NR}(A) P$; | $\quad \mid \quad A_k \leftarrow 2A_{k-1} - A_{k-1} A A_{k-1}$ |
|  | **end** |
| **Output:** $\hat{S}$ | **Output:** $A_\mathcal{T}$ |

we find empirically that random sampling is considerably sensitive to the choice of $m$. We hence explore two additional options by leveraging the structural prior of visual data: (1) Using one convolutional layer with kernel size $k$ and stride $k$ to learn $\widetilde{Q}$, and (2) Using average pooling with kernel size $k$ and stride $k$ to generate $\widetilde{Q}$. For both, we need to reshape $Q$ to the form of $\mathbb{R}^{H \times W \times d_e}$. Each slide of convolution or pooling produces a token. We set $k$ according to the length of $Q$ such that $m$ tokens can be obtained. Our experiments show that a convolution layer performs better in accuracy. We therefore use a convolution layer by default.

As $K$ is identical to $Q$, we have $\widetilde{K} = \widetilde{Q}$. Given these $m$ tokens, we then compute $A$ and $P$ as:

$$A = \exp(\widetilde{Q} \ominus \widetilde{K}), \quad P = \exp(\widetilde{Q} \ominus K). \tag{8}$$

We finally obtain the regularized self-attention matrix $\hat{S}$ of SOFT as:

$$\hat{S} = \exp\left(Q \ominus \widetilde{K}\right) \left(\exp\left(\widetilde{Q} \ominus \widetilde{K}\right)\right)^\dagger \exp\left(\widetilde{Q} \ominus K\right), \tag{9}$$

leading to Algorithm 1. The low-rank regularization is conducted as follows. For computing the attention score between any two tokens, we first correlate each of them with sampled tokens using our self-attention function (Eq (5)); With this correlation representation we then compute their similarity under the modulation of the generalized inverse of $\widetilde{Q}$'s correlation matrix. Similar as standard Nyström, our design associates the input tokens w.r.t. a small space spanned by sampled tokens, giving a proper estimation of the original attention relationships subject to a low-rank constraint. The correctness of this method is proved in Supplementary A.1.

**Moore-Penrose inverse** An accurate and commonly used way to calculate the Moore-Penrose inverse is to use Singular Value Decomposition (SVD). Given $A \in \mathbb{R}^{m \times m}$ and its SVD form $A = U\Sigma V^\top$ where $U, V$ are $m \times m$ unitary matrices and $\Sigma$ is a $m \times m$ diagonal matrix, the Moore-Penrose inverse of $A$ is $A^\dagger = V\Sigma^\dagger U^\top$. Nevertheless, SVD is not friendly to the training process on GPU hence harming the model training efficiency. To solve this issue, we adopt the Newton–Raphson method. It is an iterative algorithm with the $(k+1)$-th iteration formulated given the previous iteration as:

$$A_{k+1} = 2A_k - A_k A A_k, \quad \text{and} \quad A_0 = \alpha A. \tag{10}$$

We now prove that $A_k$ finally converges to Moore-Penrose inverse of $A_{m \times m}$, if $\alpha$ is sufficiently small [3].

**Theorem 1** *When $\alpha$ is sufficiently small, $A_{k+1} = 2A_k - A_k A A_k$, $A_k$ converges to $A^\dagger$.*

Though $\alpha = 2/\|A\|_1^2$ which ensures good convergence behavior in Algorithm 2 (see more details in Supplementary A.2.1), in practice, we find that using an alternative form gives more stable training and faster convergence. Specifically, in $\|I - A\frac{2\beta^n}{\|A\|_1^2}\|_1 \leq 1$ where $\beta$ equals to $0.5$, we find the smallest $n_i$ that holds this inequality. Then, we initialize $\alpha$ as $\alpha = \frac{2\beta^{n_i}}{\|A\|_1^2}$.

The following proposition comes with the proof of Theorem 1:

**Proposition 1** $\|AA_kA - A\|$ *and* $\|A_k - A^\dagger\|$ *decreases to 0 monotonously, if $\alpha$ is sufficiently small.*

The detail of proposition 1 is shown in Supplementary A.2.2. This ensures that our estimated inverse is sufficiently accurate for matrix decomposition, subject to that our SOFT attention is regularized.

| Methods | Complexity | Memory | Params | FLOPs | Throughput (img/s) | Top-1 % |
|---|---|---|---|---|---|---|
| Transformer [33] | $\mathcal{O}(n^2)$ | 19.0GB† | 13M | 3.9G | 1073 / 3240 | 79.1 |
| Linformer [34] | $\mathcal{O}(n)$ | 11.7GB | 13M | 1.9G | 2767 / 3779 | 78.2 |
| Performer [5] | $\mathcal{O}(n)$ | 15.0GB | 13M | 2.2G | 2037 / 3657 | 76.1 |
| Nyströmformer [39] | $\mathcal{O}(n)$ | 17.2GB | 13M | 2.0G | 1891 / 3518 | 78.6 |
| **SOFT** | $\mathcal{O}(n)$ | 15.8GB | 13M | 1.9G | 1730 / 3436 | **79.3** |

Table 1: Comparison of different linear/efficient transformer variants on ImageNet [9], based on our multi-stage Tiny configuration (see Table 2). The memory usage is measured with the batch size of 1024 which is our standard training setting. Transformer is tested at a batch size of 256, which is the maximal number possible with the GPU resource at our disposal. Throughput is in format as Train throughput / inference throughput.

**Complexity** We summarize the complexity of SOFT in space and time. For *time complexity*, it involves: (1) Sampling: $\mathcal{O}(nd_e)$. (2) Calculating three decomposed matrices: $\mathcal{O}(nmd_e + mnd_e + m^2 d_e) = \mathcal{O}(2mnd_e + m^2 d_e)$; (3) Moore-Penrose inverse: $\mathcal{O}(\mathcal{T} \times m^3) = \mathcal{O}(\mathcal{T}m^3)$, where $\mathcal{T}$ is the iteration steps. (4) All matrix multiplication: $\mathcal{O}(nm^2 + mnd_e + mnd_e) = \mathcal{O}(nm^2 + 2mnd_e)$. The total time complexity is $\mathcal{O}((d_e + 4md_e + m^2)n + \mathcal{T}m^3 + d_e m^2)$. The *space complexity* is decided by four decomposed matrices with $\mathcal{O}(n \times m) + \mathcal{O}(m \times m) + \mathcal{O}(m \times n) + \mathcal{O}(n \times d_e) = \mathcal{O}((2m + d_e)n + m^2)$. As we keep $m$ ($m \ll n$) a fixed constant in our model, both time and space complexity are $\mathcal{O}(n)$, making SOFT a linear self-attention.

### 3.3 Instantiations

Figure 2 shows how our proposed *softmax-free self-attention* block (**SOFT block**) can be implemented in a neural network. We replace the self-attention block with our SOFT block in the traditional Transformer, that is, we stack a SOFT block with a feed forward residual block [11] to form a *softmax-free Transformer* layer (**SOFT layer**).

Focusing on the general image recognition tasks, we integrate our SOFT layer into the recent pyramidal Transformer architecture [35] to form our final model **SOFT**. Further, several improvements are introduced in patch embedding (*i.e.*, tokenization). Specifically, unlike [35] that uses a combination of non-overlapping convolution and layer normalization [1], we adopt a stack of overlapping convolutions, batch normalization [14] and ReLU non-linearity. Concretely, the STEM is implemented by 3 units of 3x3 Conv→BN→ReLU, with the stride of 2, 1, 2 respectively. Then, one such unit is applied to each of three following down-sampling operations with stride of 2 in the multi-stage architecture.

The architecture hyper-parameters of SOFT are: $d$: the input channel dimension of SOFT layer. $d_e$: the embedding dimension of tokens in SOFT block. In practice, we set $d_e = d$. $h$: the head number of SOFT block. $d_h$: the channel dimension of each head and $d_h = d_e/h$. $n$: the input token sequence length of a SOFT block. $m$: the bottleneck token sequence length of SOFT block. $sp$: the sampling ratio of token sequence length sampling, which is the ratio between input token sequence length and the bottleneck token sequence length. $e$: the expansion ratio of the 2-layer feed forward block. In SOFT, for all the stages we set $d_h = 32$, $e = 4$ and $m = 49$, $sp$ varies in each stage according to the input token sequence length. Table 2 details the family of our SOFT configurations with varying capacities (depth and width).

## 4 Experiments

### 4.1 Setup

**Dataset:** We evaluate the proposed SOFT on the ILSVRC-2012 ImageNet-1K dataset [9] with 1.28M training images and 50K validation images from 1,000 classes. Following the common practice, we train a model on the training set and evaluate on the validation set. **Metrics:** For model performance, the top-1 accuracy on a single crop is reported. To assess the cost-effectiveness, we also report the model size and floating point operations (*i.e.*, FLOPs). **Implementation details:** We use the code base [37] with the default setting to train and test all the models. Specifically, we use weight decay of 0.05 and 10 epochs of linear warm-up. We conduct 300 epochs training with an AdamW optimizer

|  | Tiny | | Small | | Medium | | Large | | Huge | |
|---|---|---|---|---|---|---|---|---|---|---|
| **Stage 1** | C33-BN-ReLU, 64-d | | | | | | | | | |
| | sp. 8x8, 64-d, 2-h | x 1 | sp. 8x8, 64-d, 2-h | x 1 | sp. 8x8, 64-d, 2-h | x 1 | sp. 8x8, 64-d, 2-h | x 1 | sp. 8x8, 64-d, 2-h | x 1 |
| **Stage 2** | C31-BN-ReLU, 128-d | | | | | | | | | |
| | sp. 4x4, 128-d, 4-h | x 2 | sp. 4x4, 128-d, 4-h | x 3 | sp. 4x4, 128-d, 4-h | x 3 | sp. 4x4, 128-d, 4-h | x 3 | sp. 4x4, 128-d, 4-h | x 5 |
| **Stage 3** | C31-BN-ReLU, 320-d or 288-d | | | | | | | | | |
| | sp. 2x2, 320-d, 10-h | x 3 | sp. 2x2, 320-d, 10-h | x 7 | sp. 2x2, 288-d, 9-h | x 29 | sp. 2x2, 320-d, 10-h | x 40 | sp. 2x2, 352-d, 11-h | x 49 |
| **Stage 4 w. cls token** | C31-BN-ReLU, 512-d | | | | | | | | | |
| | sp. 1x1, 512-d, 16-h | x 2 | sp. 1x1, 512-d, 16-h | x 4 | sp. 1x1, 512-d, 16-h | x 5 | sp. 1x1, 512-d, 16-h | x 5 | sp. 1x1, 512-d, 16-h | x 5 |

Table 2: Architecture specifications of SOFT variants. *sp.*: sampling ratio. *-d*: the hidden dimension. *-h*: the number of heads in the self-attention block. *C33-BN-ReLU*: three 3x3 Conv-BN-ReLU, with the stride of 2, 1, 2 respectively. *C31-BN-ReLU*: one 3x3 Conv-BN-ReLU, with a stride of 2.

and decreasing learning rate with the cosine annealing schedule. During training, random flipping, mixup [43] and cutmix [42] are adopted for data augmentation. Label smoothing [28] is used for loss calculation. All our variants are trained with a batch size of 1024 on 32G NVIDIA V100 GPUs. We also implement our method using the Mindspore [22].

## 4.2 Comparison with existing linear Transformers

We compare our method with three existing linear Transformer models: Linformer [34], Performer [5], Nyströmformer [39] in terms of model complexity and accuracy.

Two experimental settings are adopted. Under the first setting, for all methods we use the same `Tiny` (Table 2) architecture for a fair comparison. That is, we replace the core self-attention block in SOFT with each baseline's own attention block with the rest of the architecture unchanged. Note that the *spatial reduction* module of [35] is a special case of Linformer [34]. We set the reduction ratio to be identical to ours. With the same uniform sampling idea, we replace the 1D window averaging of Nyströmformer [39] (for NLP tasks) with 2D average pooling (for images). The downsampling ratio remains identical to ours. It is also worth mentioning that there is no official code released for Reformer [18] and the local Sensitive Hash (LSH) module has strict requirements on the length of input tokens. We thus do not include this method in our comparison.

From Table 1 we can make the following observations: (i) Linear Transformer methods substantially reduce the memory and FLOPs while maintain similar parameter size comparing to the Transformer on the `Tiny` architecture; (ii) Our approach SOFT achieves the best classification accuracy among all the linearization methods. (iii) Our inference speed is on-par with other compared linear Transformers and our training speed is slightly slower than Nystromformer and both are slower than Performer and Linformer. Note that the slow training speed of our model is mostly due to the Newton-Raphson iteration which can only be applied sequentially for ensuring the accuracy of Moore-Penrose inverse. In summary, due to the on-par inference speed we consider the training cost increase is a price worth paying for our superior accuracy.

Under the second setting, we focus on the memory efficiency of SOFT against the baselines. Here we follow the ViT [11] network structure, stacking 12 attention layers with hidden dimension $d = 384$, heads $h = 12$, bottleneck token sequence length $m = 49$. Different attention blocks from the three linearized Transformer variants, Linformer [34], Performer [5], and Nyströmformer [39] are studied. For each Transformer variant, we adjust its token sequence length $n$ in a linear increment. Specifically, we use a token sequence length of $784 \times p$ where $p = 1, 2, 3, 4, 5, 6, 7, 8$ and set batch size 1 to verify whether the memory consumption increases "quadratically" or "linearly". Figure 1b shows all compared transformer variants including our SOFT indeed have a linear memory usage complexity. This is in contrast with the standard Transformer which cannot cope with long token sequences with a quadratic complexity.

## 4.3 Comparison with state-of-the-art CNNs and ViTs

| Model | Style | Resolution | M.S. Out.? | Params | FLOPs | Top-1 %. |
|---|---|---|---|---|---|---|
| ResNet-18 [13] | ConvNets | $224^2$ | ✓ | 11M | 1.9G | 69.8 |
| PVT-Tiny [35] | Transformers | $224^2$ | ✓ | 13M | 1.9G† | 75.1 |
| Coat-Lite Mini [40] | Transformers | $224^2$ | ✓ | 11M | 2.0G | 78.9 |
| LambdaNets-50 [2] | Transformers | $224^2$ | ✓ | 16M | - | 78.9 |
| **SOFT-Tiny** | **SOFT** | $224^2$ | ✓ | **13M** | **1.9G** | **79.3** |
| ResNet-50 [13] | Convolution | $224^2$ | ✓ | 25M | 4.1G | 78.5 |
| PVT-Small [35] | Transformer | $224^2$ | ✓ | 24M | 4.0G† | 79.8 |
| Deit-Small [31] | Transformer | $224^2$ | ✗ | 22M | 4.6G | 79.9 |
| T2T-ViT$_t$-14 [41] | Transformer | $224^2$ | ✗ | 21M | 5.2G | 80.7 |
| CPVT-Small [7] | Transformer | $224^2$ | ✓ | 22M | - | 79.9 |
| Twins-SVT-S [6] | Hybrid | $224^2$ | ✓ | 24M | 3.7G | 81.7 |
| **SOFT-Small** | **SOFT** | $224^2$ | ✓ | **24M** | **3.3G** | **82.2** |
| ResNet-101 [13] | Convolution | $224^2$ | ✓ | 44M | 7.9G | 79.8 |
| PVT-Medium [35] | Transformer | $224^2$ | ✓ | 44M | 7.0G† | 81.2 |
| ViT-Small/16 [11] | Transformer | $224^2$ | ✗ | 48M | 9.9G | 80.8 |
| **SOFT-Medium** | **SOFT** | $224^2$ | ✓ | **45M** | **7.2G** | **82.9** |
| ResNet-152 [13] | Convolution | $224^2$ | ✓ | 60M | 11.6G | 80.8 |
| PVT-Large [35] | Transformer | $224^2$ | ✓ | 61M | 10.1G† | 81.7 |
| T2T-ViT$_t$-24 [41] | Transformer | $224^2$ | ✗ | 64M | 13.2G | 82.2 |
| BoTNet-S1-110[27] | Hybrid | $224^2$ | ✓ | 55M | - | 82.8 |
| **SOFT-Large** | **SOFT** | $224^2$ | ✓ | **64M** | **11.0G** | **83.1** |
| CaiT-S36[32] | Transformer | $224^2$ | ✓ | 88M | 13.9G | 83.3 |
| Swin-B[20] | Transformer | $224^2$ | ✓ | 88M | 15.4G | 83.3 |
| Twins-SVT-L [6] | Hybrid | $224^2$ | ✓ | 99M | 14.8G | 83.3 |
| **SOFT-Huge** | **SOFT** | $224^2$ | ✓ | **87M** | **16.3G** | **83.3** |

Table 3: Evaluation results on ILSVRC-2012 ImageNet-1K [9] `validation` set. We report the results using the input size of 224x224 pixels center cropped from resized images with 256x256 pixels. `M.S.Out.` stands for whether the model is designed for multi-scale output. †: Corrected FLOPs by taking into account the cost of attention matrix multiplication overlooked in the origin paper.

We compare with state-of-the-art alternatives and report the top-1 accuracy on the ImageNet-1K validation set. FLOPs are calculated at batch size 1024. From Figure 1a and Table 3, the following observations are made: (i) Overall, ViT and its variants yield better classification accuracy over CNNs. (ii) We achieve the best performance among the recent pure vision Transformer based methods including ViT [11] and DeiT [31], as well as the state-of-the-art CNN RegNet [26]. (iii) Our SOFT outperforms the most similar (in architecture configuration) Transformer counterparts PVT [35] at all variants. Since the attention module is the main difference, this validates directly the effectiveness of our model. (iv) We can also beat the latest ViT variants Twins [6] which is designed to address the efficiency limitation of ViT. We have done so with less parameters and fewer float point computation.

To gain some insights into how attention is learned using our SOFT and the alternatives, Figure 3 shows the attention masks of various compared models. For each model, we show the output from the first two attention heads. It is evident that SOFT exhibits robustness and versatility in capturing local and long distance relations among pixels. It is interesting to note that, although SOFT is trained on an object categorization dataset in ImageNet [9], it seems to be able to learn both semantic concepts shared across instances in the same category and instance specific features. For instance, in the bottom-right example of a bird class, one attention head focuses on the black bird only, while the other attend to both birds in the image. More examples are shown in Supplementary A.4.

## 4.4 Ablation studies

**Pyramidal architecture:** Unlike the earlier non-pyramidal vision Transformers (*e.g.*, ViT [11]), most recent pyramidal (multi-scale) Transformers (*e.g.*, PVT [35]) use convolution layers to reduce

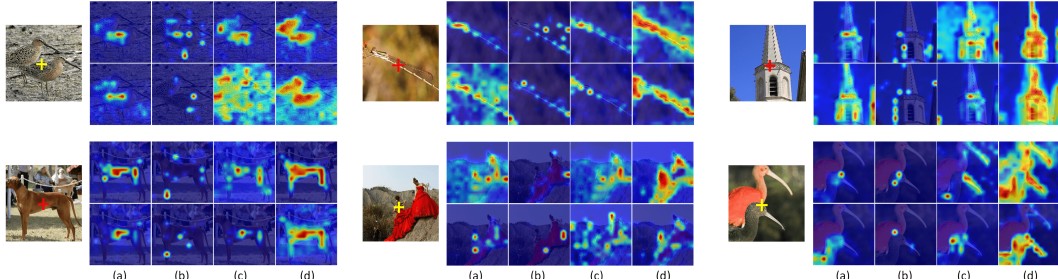

Figure 3: Comparing the attention heatmaps of a query patch (marked by the cross "+") against all the patches of an image, produced by (a) Transformer [33], (b) Performer [5], (c) Nystromformer [39] and (d) Our SOFT. See Supplementary A.4 for more examples.

the spatial resolution (*i.e.*, token sequence length) between stages. In this study, we ablate SOFT with a pyramidal architecture (our default SOFT-Small), SOFT w/o a pyramidal architecture and DeiT-S [31] (no pyramidal architecture either). We replace the Transformer layer with a SOFT layer to get SOFT w/o a pyramidal architecture. Note all three variants have similar parameters and FLOPs. Table 5a shows that the conv-based pyramidal architecture is clearly superior to a non-pyramidal design, and our non-pyramidal counterpart is even slightly better than DeiT-S [31] whilst enjoying linear complexity.

**Bottleneck token sequence length:** In this study, we examine how the bottleneck token sequence length $m$, sampled from $n$ tokens, influences the model's performance. We change the bottleneck token sequence length in all stages to $36, 49, 64, 81$. Table 4a shows that longer bottleneck token would increase the memory cost and the computational overhead. $m = 49$ seems to give the best trade-off between the performance and computational overhead. The memory usage is measured with the batch size of 128.

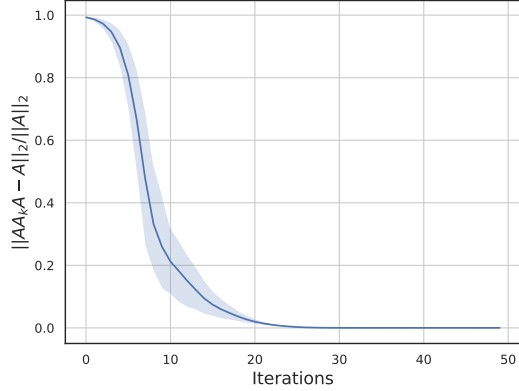

Figure 4: Convergence analysis for the approximation of the Moore-Penrose inverse on SOFT-Tiny.

**Token sampling:** The sampling function in SOFT can assume different forms. **Convolution:** The sequence $Q \in \mathbb{R}^{n \times d_e}$ is first reshaped to a feature map $\mathbb{R}^{H \times W \times d_e}$. $r \times r$ convolution kernel with stride of $r$ is applied for downsampling, where $r = \sqrt{sp}$. The output channel size is also kept and no bias is used. At last, the feature map is reshaped back to the sequence. **Average pooling:** using a $r \times r$ kernel and $r$ stride, where $r = \sqrt{sp}$. **Random sampling:** $m$ tokens are randomly picked from $n$ tokens. **Biased sampling:** We pick $m$ tokens with a biased policy. Here, the first $m$ tokens are picked. Table 4b shows that average pooling yields the best performance while with less computational overhead comparing to convolution. Biased sampling can miss the most salient samples, and there is no guarantee that random sampling can keep the uniformity of the chosen samples. This result thus justifies the choice of using average pooling in SOFT.

| Bottleneck | Memory | FLOPs | Top-1 % |
|---|---|---|---|
| 36 | 15.1GB | 1.9G | 79.0 |
| 49 | 15.8GB | 1.9G | 79.3 |
| 64 | 16.9GB | 2.0G | 79.3 |
| 81 | 18.5GB | 2.1G | 78.9 |

| Sampling methods | Params | FLOPs | Top-1 % |
|---|---|---|---|
| Convolution | 13.07M | 2.0G | 79.3 |
| Random sampling | 12.96M | 1.9G | 79.3 |
| Biased sampling | 12.96M | 1.9G | 79.0 |
| Average pooling | 12.96M | 1.9G | 79.3 |

Table 4: (a) Ablations on bottleneck token sequence length. (b) Ablations on sampling methods.

**Overlapped convolution:** We ablate SOFT with overlapped convolution (our default choice, same as many recent works) and SOFT with non-overlapped convolution in our `Tiny` configuration. Table 5b shows that SOFT with overlapped convolution performs better than SOFT without overlapped convolution. Our non-overlapped convolution variant still outperforms the PVT [35] which also has the same non-overlapped convolution by a clear margin.

| Methods | Pyramidal? | Top-1 % | Methods | Overlapped? | Top-1 % |
|---------|:----------:|:-------:|---------|:-----------:|:-------:|
| DeiT-S [31] | ✗ | 79.8 | PVT [35] | ✗ | 75.1 |
| SOFT | ✗ | 80.1 | SOFT | ✗ | 77.4 |
| SOFT | ✓ | 82.2 | SOFT | ✓ | 79.3 |

Table 5: (a) Ablations on pyramidal architecture. (b) Ablations on overlapped convolution.

**Newton-Raphson's convergence:** We study how many iterations the Newton-Raphson method needs to converge when computing the Moore-Penrose inverse $A^\dagger$. We use $\|AA_kA - A\|_p/\|A\|_p$ with $p = 2$ (see Proposition 1) as the convergence metric to quantify the difference between $A_k$ and $A^\dagger$. Figure 4 shows that our approximation converges within 20 iterations across all stages.

## 4.5 Additional experiments on NLP tasks

In this section, we evaluate our method against other linear counterparts on four tasks of the Long Range Arena (LRA) [29] benchmark *e.g.*, Listops [23], byte-level IMDb reviews text classification [21], byte-level document retrieval [25], and image classification on sequences of pixels [19].

**Implementations.** We use the Pytorch version of LRA [29] benchmark, provided by [39]. For the evaluation protocol, we strictly follow [29, 39]. We omit the Pathfinder(1K) task as we cannot replicate the result of Nyströmformer [39]. For our SOFT, we simply use the average pooling with window size 4, stride 4 to sample the bottlenecks. We follow the configurations of [39], with 2 layers, 64 and 128 hidden dimension respectively, and 2 attention heads. The results in Table 6 shows that our SOFT outperforms both the standard and alternative efficient Transformers on three out of four tasks, as well as the average result.

| Methods | Listops(2K) | Text(4K) | Retrieval(4K) | Image(1K) | Avg. % |
|---------|:-----------:|:--------:|:-------------:|:---------:|:------:|
| Transformer [33] | 37.10 | 65.02 | 79.35 | 38.20 | 54.92 |
| Reformer [18] | 19.05 | 64.88 | 78.64 | 43.29 | 51.47 |
| Linformer [34] | 37.25 | 55.91 | 79.37 | 37.84 | 52.59 |
| Performer [5] | 18.80 | 63.81 | 78.62 | 37.07 | 49.58 |
| Nyströmformer [39] | 37.15 | **65.52** | 79.56 | 41.58 | 55.95 |
| **SOFT** | **37.40** | 63.49 | **81.77** | **46.91** | **57.39** |

Table 6: Comparison of different linear/efficient Transformer variants on Long Range Arena [29], based on its official configuration. Our SOFT surpasses previous efficient methods on three tasks.

## 5 Conclusions

We have introduced a novel softmax-free self-attention (SOFT) mechanism for linearizing Transformer's complexity in space and time. Unlike existing linear Transformers that aim to approximate the conventional softmax based self-attention, SOFT employs a Gaussian kernel based attention which eliminates the need for softmax normalization. This design enables a full self-attention matrix to be approximated via a low-rank matrix decomposition. The robustness of the approximation is achieved by calculating its Moore-Penrose inverse using a Newton-Raphson method. Extensive experiments show that SOFT yields superior trade-off in accuracy and complexity.

**Acknowledgment** This work was funded in part by Shanghai Municipal Science and Technology Major Projects (No.2018SHZDZX01 and No.2021SHZDZX0103), Mindspore, National Science Foundation of China under Grant No.11690013, 71991471 and the scientific-technological innovation plan program of Universities guided by the Ministry of Education, China.

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
