# Supplementary for SOFT: Softmax-free Transformer with Linear Complexity

**Jiachen Lu**[1]  **Jinghan Yao**[1]  **Junge Zhang**[1]  **Xiatian Zhu**[2]  **Hang Xu**[3]
**Weiguo Gao**[1]  **Chunjing Xu**[3]  **Tao Xiang**[2]  **Li Zhang**[1][*]

[1]Fudan University     [2]University of Surrey     [3]Huawei Noah's Ark Lab

https://fudan-zvg.github.io/SOFT

## A  Appendix

### A.1  Nyström method

Nyström method [5] aims to calculate a low-rank approximation for a Gram matrix. For Transformers, the self-attention matrix can be viewed as a Gram matrix $S$ with a Gaussian kernel $k$ applied to the query $Q$, with each element $S_{ij}$ expressed as:

$$S_{ij} = k\big(Q_{i,:}, Q_{j,:}\big) = \exp(-\frac{\|Q_{i,:} - Q_{j,:}\|_2^2}{2\sqrt{d}}), \tag{1}$$

$k(x, y)$ means operating Gaussian kernel $k$ to $(x, y)$, which can be written in the feature space as:

$$k(x, y) = \sum_{i=1}^{n} \lambda_i \phi_i(x) \phi_i(y), \tag{2}$$

$n$ is the dimension of a feature space, $\lambda_i$ denotes the eigenvalue and $\phi_i$ denotes the eigenfunction of kernel $k$. According to the eigenfunction's definition, we can get:

$$\int k(y, x) \phi_i(x) p(x) dx = \lambda_i \phi_i(y), \tag{3}$$

where $p(x)$ is the probability distribution of $x$. And $\{\phi_i(x)\}$ are $p$-orthogonal:

$$\int \phi_i(x) \phi_j(x) p(x) dx = \delta_{ij}. \tag{4}$$

$\delta_{ij}$ is 0 when $i \neq j$, 1 when $i = j$. To get an approximation of the eigenfunctions, we sample $\{x_1, x_2, \cdots, x_q\}$ from $p(x)$, then:

$$\frac{1}{q} \sum_{t=1}^{q} k(y, x_t) \phi_i(x_t) \approx \lambda_i \phi_i(y), \tag{5}$$

$$\frac{1}{q} \sum_{t=1}^{q} \phi_i(x_t) \phi_j(x_t) \approx \delta_{ij}. \tag{6}$$

This inspires us to approximate the Gram matrix $S$. Let $S^{(m)}$ be a submatrix of $S$, consisting of $m \times m$ elements from $S$. Gram matrix is a symmetric positive semi-definite matrix, so it has a spectral decomposition:

$$S^{(m)} U^{(m)} = U^{(m)} \Lambda^{(m)}, \tag{7}$$

---
[*]Li Zhang (lizhangfd@fudan.edu.cn) is the corresponding author with School of Data Science, Fudan University.

35th Conference on Neural Information Processing Systems (NeurIPS 2021).

where $U^{(m)}$ is column orthogonal and $\Lambda^{(m)}$ is a diagonal matrix with the diagonal elements as the eigenvalues of $S^{(m)}$. Substituting the $y$ to $x_j$ and applying the approximation above to $S$, we can get:

$$\phi_i(x_j) \approx \sqrt{m}U_{j,i}^{(m)}, \quad \lambda_i \approx \frac{\lambda_i^{(m)}}{m}, \tag{8}$$

$$\phi_i(y) \approx \frac{\sqrt{m}}{\lambda_i^{(m)}} \sum_{t=1}^{m} k(y, x_t)\phi_i(x_t), \tag{9}$$

$\lambda_i$ is eigenvalue of $S$ and $\lambda_i^{(m)}$ is the eigenvalue of $S^{(m)}$. Denote $\tilde{S}$ as the rank-$m$ approximation of $S$ and $\tilde{U}, \tilde{\Lambda}$ as the approximation for spectral decomposition of $S$. Now we can get an approximation of $S$ with rank $m$:

$$\tilde{S} = \tilde{U}\tilde{\Lambda}\tilde{U}^T = \sum_{t=1}^{m} \tilde{\lambda}_t^{(n)} \tilde{u}_t^{(n)}(\tilde{u}_t^{(n)})^T. \tag{10}$$

Similarly, we have:

$$\phi_i(x_j) \approx \sqrt{n}U_{j,i}(n), \quad \lambda_i \approx \frac{\tilde{\lambda}_i^{(n)}}{n}. \tag{11}$$

Thus

$$\tilde{\lambda}_i^{(n)} \approx \frac{n\lambda_i^{(m)}}{m}, \tag{12}$$

$$\tilde{u}_t^{(n)} \approx \sqrt{\frac{m}{n}} \frac{1}{\lambda_t^{(m)}} S_{n,m}u_t^{(m)}. \tag{13}$$

Then we get an approximation of $S$: $\tilde{S} \approx S_{n,m}S_{m,m}^{\dagger}S_{m,n}$. $S$ has a block representation below:

$$S = \begin{bmatrix} S_{m,m} & S_{m,n-m} \\ S_{n-m,m} & S_{n-m,n-m} \end{bmatrix}. \tag{14}$$

## A.2 Newton method

### A.2.1 Proof of theorem 1

**Proof A.1** *A is a symmetric positive semi-definite matrix and $A_{ij} \leq 1$, $\forall 1 \leq i, j \leq n$, $A_{ii} = 1$, $1 \leq i \leq n$ in our case. $A_0$ is chosen to be $\alpha A$, so the $A_k$ can be written as $A_k = C_k A = AD_k$ for some matrix $C_k, D_k$, leading to the fact that*

$$A^{\dagger}AA_k = A_k, \quad A_k AA^{\dagger} = A_k. \tag{15}$$

*This is because $A_{k+1} = A_k(2I_n - AA_k) = (2I_n - A_kA)A_k$ and $A_0 = \alpha A$. We make a difference between $A^{\dagger}$ and $A_{k+1}$:*

$$\begin{aligned} A^{\dagger} - A_{k+1} &= A^{\dagger} - 2A_k + A_kAA_k \\ &= A^{\dagger} - A_kAA^{\dagger} - A^{\dagger}AA_k + A_kAA_k \\ &= (A^{\dagger} - A_k)A(A^{\dagger} - A_k). \end{aligned} \tag{16}$$

*We norm both sides of the equation above:*

$$\begin{aligned} \|A^{\dagger} - A_{k+1}\| &= \|(A^{\dagger} - A_k)A(A^{\dagger} - A_k)\| \\ &\leq \|A^{\dagger} - A_k\|\|A(A^{\dagger} - A_k)\|. \end{aligned} \tag{17}$$

*And we left multiply $A$ on the both sides of (16), then norm the equation:*

$$\begin{aligned} \|AA^{\dagger} - AA_{k+1}\| &= \|A(A^{\dagger} - A_k)A(A^{\dagger} - A_k)\| \\ &\leq \|AA^{\dagger} - AA_k\|^2. \end{aligned} \tag{18}$$

*We choose $\alpha$ sufficiently small so that the initial value satisfy $\|AA^{\dagger} - AA_0\| < 1$. We set $\alpha = \frac{2}{\|A\|_1^2}$ to ensure it is small enough [1]. Then the $\|AA^{\dagger} - AA_k\| \to 0$, when $k \to \infty$. The inequality (17) implies that $A_k \to A^{\dagger}$, $k \to \infty$.*

### A.2.2 Proof of proposition 1

**Proof A.2** *Note that when we multiply $A$ on both sides of (16), the equation turns to be:*

$$A - AA_{k+1}A = A(A^\dagger - A_k)A(A^\dagger - A_k)A$$
$$= (AA^\dagger - AA_k)(A - AA_kA). \tag{19}$$

*Similarly norm both sides of (19), considering that $\|AA^\dagger - AA_k\| \to 0$ and $\|AA^\dagger - AA_k\| < 1$ always holds, $\|A - AA_kA\|$ monotonically decreases to $0$. The inequality (17) implies that $\|A_k - A^\dagger\|$ decreases to $0$ monotonously.*

Note that although $\|A - AA_kA\|$ monotonically decreases to $0$, $\|A_kAA_k - A_k\|$ cannot be proved to monotonically decrease to $0$.

### A.3 Non-linearized gaussian kernel attention

In our formulation, instead of directly calculating the Gaussian kernel weights, they are approximated. More specifically, the relation between any two tokens is reconstructed via sampled bottleneck tokens. As the number $m$ (e.g., 49), of bottleneck tokens is much smaller than the token sequence length, our attention matrix is of low-rank. This has two favorable consequences: **(I)** The model now focuses the attentive learning on latent salient information captured by the $m$ bottleneck tokens. **(II)** The model becomes more robust against the underlying token noise due to the auto-encoder style reconstruction [3].

This explains why the model with an approximated gram matrix performs better than the one with a directly estimated matrix. Further, exact Gaussian kernel attention computation leads to training difficulties. We first hypothesized that this might be due to lacking normalization (as normalization often helps with training stability and convergence), and tested a variant with softmax on top of an exact Gaussian kernel attention matrix. However, it turns out to suffer from a similar failure. We cannot find a solid hypothesis so far and will keep investigate this problem.

### A.4 Attention visualization

Figure 1 shows more visualization of the attention masks by various Transformers [4, 2, 6] and our SOFT. For each model, we show the output from the first two attention heads (up and down row). It is noteworthy that SOFT exhibits better semantic diversity of the multi-head mechanism than other methods. Moreover, when we sample the patch at the boundary of multiple objects, SOFT is able to more precisely capture all these objects.

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

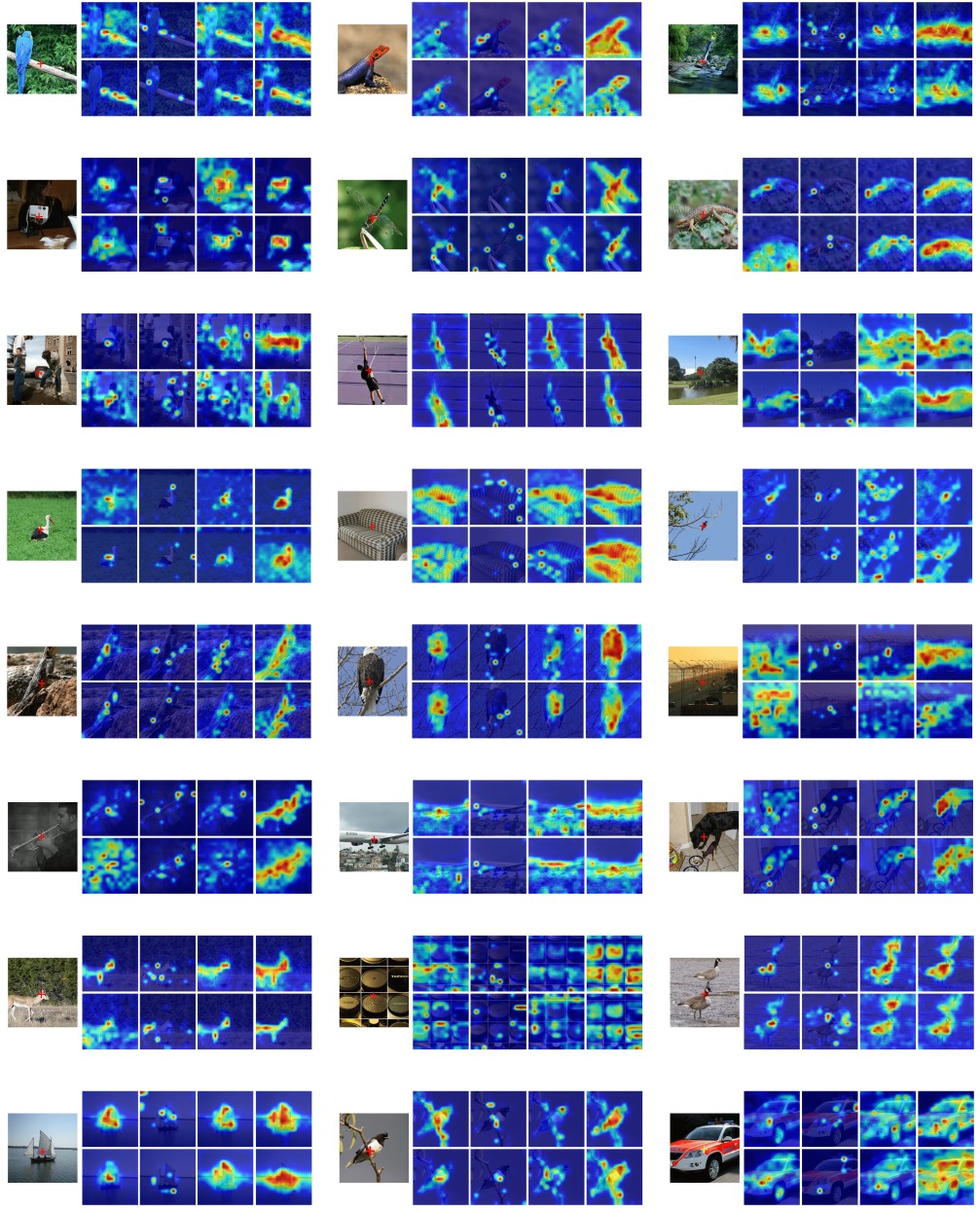

Figure 1: Comparing the attention heatmaps of a query patch (marked by the cross "+") against all the patches of an image, produced by (a) Transformer [4], (b) Performer [2], (c) Nyströmformer [6] and (d) Our SOFT.