# OpenReview forum: "SOFT: Softmax-free Transformer with Linear Complexity"
_NeurIPS.cc/2021/Conference — NeurIPS 2021 Spotlight_

### Official Review · Reviewer_T2zb · 2021-07-15

**Rating:** 7
**Confidence:** 4

**Summary:**

The paper substitutes softmax self-attention with a Gaussian kernel function. This allows to approximate the self-attention matrix with a low-rank approximation. The robustness of the approximation is supported by theoretical guarantees, while previous works that don't provide these guarantees. The proposed method is evaluated on ImageNet, arguably the most popular benchmark for evaluating computer vision models, and shows impressive results when compared against other Linear Transformer alternatives and even when compared against the standard (full) self-attention approach.

**Limitations And Societal Impact:**

The paper does not discuss any potential negative societal impacts of the work. The paper is more on the theory side of the theory-application spectrum of papers, but it targets (and tests the proposed method on) an image recognition task, i.e. ImageNet. In any case, I do not think that this work raises specific concerns that are not present in a large fraction of NeurIPS submissions, or in virtually all publications applying Machine Learning to Computer Vision.

**Main Review:**

=== UPDATE AFTER DISCUSSION PERIOD ===

In addition to the strengths below, the auhors have clarified my questions and addressed satisfactorily most of my concerns. Thus, I'm advocating for accepting the paper (I'm increasing my score from 6 to 7).

=== Strengths ===

1) The paper addresses an important topic, i.e. efficient self-attention mechanisms, with numerous applications in computer vision, and natural language processing.

2) The strong results shown on ImageNet accuracy, compared to other popular efficient attention alternatives, makes it a good candidate to be useful to practitioners and influential to future research.

3) The method presented in the paper is built from strong theoretical foundations (assuming the proof of Theorem 1 is correct, see clarifications below).

4) The paper carefully analyses the problems in many existing "efficient" (Linear) transformer alternatives. Mainly focusing on Nyströmformer which is the most similar existing approach.

5) The experimental details are well detailed in section 4.1, which makes easier for other to build on and compare with the results of the paper.

=== Weaknesses ===

1) I find the definition of $n$ a little bit confusing in some parts of the text. $n$ is first defined as the "sequence length" (line 94), but then is referred to as "token length" (Figure 1b and sections 3.3 and 4.2).

2) Assuming that in all cases $n$ is referring to the same concept (i.e. sequence length, or number of tokens per image), then some of the experiments don't make much sense. In particular the experiments where SOFT is used with the VIT architecture (i.e. Figure 1b). Lengths of 784 x p (with p=1...8) are evaluated, (in all cases using m=49?) but these lengths are unreasonable since the VIT paper uses patches of 32x32 or 16x16, which results in inputs of 49 or 196 tokens in total to the attention layer.

3) In general, I think that a plot/table showing the optimal m (w.r.t. accuracy/time or accuracy/flops) for different sequence lengths should be reported. This is important because the value of m chosen is 49, which is already "large" for typical non-pyramidal transformers. Thus, in this regime it might be that the proposed method does not provide any significant speed-up. An alternative experiment would be to use the PVT architecture with different input resolutions (but same downscaling factors in each stage): e.g. 150,224,280,380,448,...

4) Figure 1a shows accuracy vs. # of paramaters vs. accuracy. This plot seems of little relevance for the goal of this work. A plot showing accuracy vs. training/inference flops; or even better accuracy vs. training/inference time would be preferable. Accuracy and flops are reported in Table 3, but given that the proposed method uses an iterative algorithm to approximate the pseudo-inverse, and iterative algorithms are not ideal for modern accelerators (GPUs/TPUs), I would strongly recommend the authors to report runtime.

5) One of the limitations of the presented of method, as described in lines 115-117, is that "we find that when using our kernel-based self-attention matrix $S$ without linearization, the training of a transformer fails to converge". Do you have an hypothesis that explains why this happens?

6) All the experiments are limited to ImageNet recognition. It's unclear whether they run a single run for each presented result or the reported numbers are the average of multiple independent runs. In any case, I would recommend the authors to perform additional experiments (if time permits) on a different benchmark. For instance, the benchmark presented in the paper "Long Range Arena: A Benchmark for Efficient Transformers" (https://arxiv.org/pdf/2011.04006.pdf), which also highlights that existing "efficient" attention mechanisms are far from perfect, seems like a good benchmark to evaluate the proposed method.

=== Clarifications ===

I can't follow the proof of Theorem 1. In particular going from Eq. (18) in the appendix to Eq. (19).  Eq. (18) reads:
$
||A^\dagger - A_{k+1}|| =  ||(A^\dagger - A_{k})A(A^\dagger - A_{k})||
$

Line 36 reads: "And we left multiply A on the both sides of (17), then norm the equation". Then Eq. (19) reads:
$
||A A^\dagger - A A_{k+1}|| =  ||A(A^\dagger - A_{k})A(A^\dagger - A_{k})||
$

This seems to imply that if $A$ is a symmetric semi-definite positve matrix and $||X|| = ||Y||$, then $||A X|| = ||A Y||$. However, this is false in general. See a counter example: X = [[1, 2], [3, 4]], Y = [[1, 3], [2, 4]], A = [[2, 1], [1, 2]] (apologies, it seems that matrices are not rendered well). $A$ is positive definite.
$||X|| = ||Y|| \approx 5.4649857$ but
$||A X|| \approx 15.42234 \not\approx 16.18217 \approx ||A Y||$.

I tried to use the fact that $||A B|| \leq ||A||\cdot||B||$ (in the case of matrix $p$-norms), but didn't get too far. Given that $||A|| > 0$ if $A$ is semi-definite positive:

$
||A^\dagger - A_{k+1}|| =  ||(A^\dagger - A_{k})A(A^\dagger - A_{k})|| \Leftrightarrow
||A|| \cdot ||A^\dagger - A_{k+1}|| =  ||A||\cdot||(A^\dagger - A_{k})A(A^\dagger - A_{k})||
$

Then:

$
||AA^\dagger - AA_{k+1}|| \leq ||A|| \cdot ||A^\dagger - A_{k+1}|| \text{  ???  }
||A(A^\dagger - A_{k})A(A^\dagger - A_{k})||
\leq ||A||\cdot||(A^\dagger - A_{k})A(A^\dagger - A_{k})||
$

To summarize, I need additional indications from the authors to follow the proof.

=== Other comments ===

 Please, carefully review the paper looking for typos. Some that I found:
- Main text, line 45: "identity" -> "identify".
- Main text, line 200: "AadmW" -> "AdamW".
- Appendix, equation 19: The first line ends with $|$ to denote the norm, it should end with $||$.
- Appendix, line 41: "multiple" -> "multiply"

Suggestion: $\times$ is typically used to denote the cross-product, I would recommend using $\cdot$ for denoting the product of a scalar by a matrix (Eq. 3, 4, 5) to avoid confusion, but feel free to ignore this suggestion if you strongly prefer otherwise.

**Time Spent Reviewing:**

9

---

> ### Author Response · Authors · 2021-08-10
> **Response for Reviewer T2zb**
>
> We thank the reviewer for the positive and detailed review as well as the suggestions for improvement. Our response to the reviewer’s comments is below:
>
> **Q1: Inconsistency for token sequence length.**
>
> Thanks. Both are short for “token sequence length”. We will make the terminology more consistent.
>
> **Q2: Confusion of using $784\times p$ as length.**
>
> Sorry for the confusion. As stated in L221, `m` here denotes the bottleneck (sampled) token length. We set `m`=49 in all the stages, which is independent of the token sequence length `n` (please also see our response to Q1 of Reviewer #M8ep). Secondly, the performance gain of linearization methods becomes more pronounced as the token sequence length increases. For example, the GPU memory consumption for calculating the attention matrix of a $56 \times 56$ feature map (common in pyramidal networks, which has 3136 patches, between $784 \times 3$ and $784 \times 4$) is 3 times higher compared to our SOFT.  We, therefore, ran experiments with a “linearly” increased input token length $784 \times p$ (with p=1...8) and reported the results in Figure 1b **only** to verify whether the memory consumption increases “quadratically” or “linearly” for different compared models. Our input resolution is still set to standard $224 \times 224$, with the token length of $56 \times 56$, $28 \times 28$, $14 \times 14$ and $7 \times 7$ in each stage respectively. We will clarify in the revision.
>
> **Q3: Optimal `m`.**
>
> Thanks. As explained in our response to Q2, `m`=49 refers to the sampled bottleneck token length, which in our case, denotes a $7\times7$ sized feature map rather than a $49 \times 49$ sized feature map. So it is actually quite a moderate size. In Table 4a (main paper), we also showed the performance and memory usage of SOFT with different `m` values (eg, 36, 49, 64, and 81). From the results,  we drew the conclusion that 49 is close to the optimal number for a pyramidal structure. We could run a grid search to find an even better `m` value, but that is too time-consuming for the short rebuttal period.
>
> **Q4: Training and inference time.**
>
> Good suggestion. As suggested we have now evaluated the throughput (images/sec) of our tiny model for training and inference on a machine with 8 Tesla V100 GPUs. The results are:
>
> |Methods		|Train Throughput (img/s)  |	Inference Throughput (img/s)
>   |-----------------|:---------------------:|:----------------------:|
> |Transformer		|1073		|		3240
> |Linformer		|2767		|		3779
> |Performer		|2037		|		3657
> |Nystromformer	|1891		|		3518
> |SOFT (ours)		|1730		|		3436
>
>
> From the results, it can be seen that: (1) Our inference speed is on-par with other compared linear Transformers and (2) our training speed is slightly slower than Nystromformer and both are slower than Performer and Linformer. Note that the slow training speed of our model is mostly due to the fact that no official optimized implementation of Gaussian kernel has yet been made available in Pytorch. It thus can be overcome in the future. So in summary, due to the on-par inference speed we consider the training cost increase is a price worth paying for our superior accuracy.
>
> **Q5: Hypothesis for the failure of non-linearized gaussian kernel self-attention.**
>
> Great question. In our formulation, instead of directly calculating the Gaussian kernel weights, they are approximated. More specifically, the relation between any two tokens is reconstructed via sampled bottleneck tokens. As the number `m` (e.g., 49)  of bottleneck tokens is much smaller than the token sequence length, our attention matrix is of low-rank. This has two favorable consequences:
> 1. The model now focuses the attentive learning on latent salient information captured by the `m` bottleneck tokens.
> 2. The model becomes more robust against the underlying token noise due to the auto-encoder style reconstruction.
>
> This explains why the model with an approximated gram matrix performs better than the one with a directly estimated matrix. Further, exact Gaussian kernel attention computation leads to training difficulties. We first hypothesized that this might be due to lacking normalization (as normalization often helps with training stability and convergence), and tested a variant with softmax on top of an exact Gaussian kernel attention matrix. However, it turns out to suffer from a similar failure. In the short rebuttal period we cannot find a solid hypothesis and will keep investigating this problem. Note that in the submission experimental results were included to verify how SOFT works better (see Table 4, main paper), including results obtained by varying the sampling ratio and the sampling strategy.
>
>
> **Q6: Single run or multiple runs on ImageNet? Evaluate on Long-Range-Arena benchmark.**
>
> Thanks. Following the standard protocol, the reported results on ImageNet recognition were of one run rather than average over multiple runs. This is because, given the scale of the ImageNet dataset, there is typically little performance discrepancy over multiple runs. To verify that for our model, we have now run multiple times and found that there is less than 0.01% in the performance differences over runs.
>
> As suggested, we have now also additionally tested the Long-Range-Arena benchmark. In particular, we ran experiments on five tasks: ListOps(2K), Text(4K), Retrieval(4K), Pathfinder(1K) and Image(1K). To evaluate SOFT on LRA, we used the LRA test code provided by the official Nystromformer GitHub repository and kept the settings for this benchmark with the attention module replaced with ours. We first reproduced the results of Nystromformer on each task except the Pathfinder. (For Pathfinder(1K) we cannot reproduce the result of Nystromformer and more time is needed to investigate; Now both Nystromformer and our SOFT cannot get reasonable results). The table below shows that our SOFT outperforms both the standard and alternative efficient Transformers on three out of four tasks, as well as the average result. We will add this experiment in the revised version.
>
>
>  |  Methods       |      ListOps(2K)  |Text(4K)  | Retrieval(4K)  |  Image(1K) 	|  Avg.
>  |-----------------|:---------------------:|:----------------------:|:----------:|:----------------------:|----------:|
> Transformer	|  37.10    |      65.02     |   79.35        |           38.20       |          54.92
> Reformer        |  19.05      |    64.88   |     78.64       |            43.29         |        51.47
> Linformer       |   37.25     |     55.91   |     79.37        |           37.84         |       52.59
> Performer         |18.80     |     63.81   |     78.62      |             37.07         |       49.58
> Nystromformer  |37.15    |  **65.52** |     79.56       |            41.58         |       55.95
> **SOFT**            |    **37.40**     |    63.49    |    **81.77**       |       **46.91**      |     **57.39**
>
>
> **Q7: Clarification for Theorem 1:**
>
> Sorry for the confusion. As stated in the paper, Eq. (19) actually follows Eq. (17):
>
> $A^{\dagger}-A_{k+1}=(A^{\dagger}-A_{k})A(A^{\dagger}-A_{k})$
>
> We then left multiply $A$ and norm both sides of the equation as:
>
> $|| AA^{\dagger}-AA_{k+1}||=||A(A^{\dagger}-A_k)A(A^{\dagger}-A_k)||$
>
> Therefore, Eq. (19) is based on an equation of matrix, instead of norm.
>
> **Q8: Typos and \'$\times$\' to \'$\cdot$\'.**
>
> Thanks for pointing them out. We will revise them in the revision.

---

> > ### Comment · Reviewer_T2zb · 2021-08-30
> > **Thanks for addressing all my questions**
> >
> > Thank you so much for addressing my questions with a great level of detail. I will update my review and recommend the acceptance of the paper.
> >
> > I'm looking forward to follow-up papers exploring hypothesis on why "self-attention matrix $S$ without linearization fails to converge".

---

### Official Review · Reviewer_aC3r · 2021-07-16

**Rating:** 7
**Confidence:** 3

**Summary:**

This paper proposes a novel linear complexity self-attention module that
(1)	first replaces the softmax attention with the Gaussian kernel
(2)	then apply Nystrom approximation to calculate the gram matrix with O(n) complexity.
(3)	whose bottleneck samples are calculated using average pooling or convolutional layer.

The proposed method demonstrates good top-1 accuracy on ImageNet classification with low space and time computational complexity.

**Limitations And Societal Impact:**

Limitation of the gaussian weight is briefly described in the remarks paragraph in L111~117.
There are no discussion aout the potential negative social impact. However, since it is the algorithmic paper, the possibility that the paper causes the negative impact may small.

**Main Review:**

Strength

The paper proposes a novel linear complexity self-attention architecture that applies kernel approximation technique to the attention weight.
-	Considers the attention weight calculated by Gaussian kernel.
-	Approximate the gram matrix using Nystrom method.
-	Propose a bottleneck sampling scheme using average pooling and convolutional layer.
-	Propose to calculate the inverse matrix of the gram matrix w.r.t. bottleneck samples using Newon-Raphson method, which is suitable for GPU calculation.

The paper experimentally compares the proposed method with several self-attention with linear complexity. The proposed method demonstrates both good accuracy and low computation complexity.

The proposed method demonstrates comparable accuracy to the state-of-the-art methods on ImageNet dataset.

The author conducts several ablation study with respect to the size of bottleneck samples, sampling methods, and the number of iteration for Newton-Raphson method.


Weakness

In the remarks, the author says that the model with Gaussian kernel weight without does not work well. Since the proposed method is derived as the approximation about the Gaussian kernel, I would like to see the discussion why the true weight does not work well while approximated weight shows good performance.
Regarding this, I would like to see the approximation performance of the gram matrix with the proposed approximation.

It looks like Nystromformer is the method that tries to approximate the true softmax attention. Since softmax weight works well while non-approximated Gaussian weight does not work well, I would like to know the reason of this reversal in the performance after the approximation.


Another Comments

In Table (4) Random Sampling works the same performance as Average Pooling. Though the author prefers average pooling, I personally think random sampling is better since random sampling does not violate the permutation equivariance property of the whole the layer while average pooling depends on the position information.

The proposed Gaussian weighted self-attention may be regarded as the variant of kernel density estimation.


Evaluation

Considers the theoretical and empirical performance of the proposed method, I recommend acceptance.


**Time Spent Reviewing:**

6

---

> ### Author Response · Authors · 2021-08-10
> **Response to Reviewer aC3r**
>
> We thank the reviewer for the positive and detailed review as well as the suggestions for improvement. Our response to the reviewer’s comments is below:
>
> **Q1: Explanation for the better performance of the approximated Gaussian kernel.**
>
> Great question. In our formulation, instead of directly calculating the Gaussian kernel weights, they are approximated. More specifically, the relation between any two tokens is reconstructed via sampled bottleneck tokens. As the number `m` (e.g., 49)  of bottleneck tokens is much smaller than the token sequence length, our attention matrix is of low-rank. This has two favorable consequences :
>
> 1. The model now focuses the attentive learning on latent salient information captured by the `m` bottleneck tokens;
> 2. The model becomes more robust against the underlying token noise due to the auto-encoder style reconstruction.
>
> This explains why the model with an approximated gram matrix performs better than the one with a directly estimated matrix. Further, exact Gaussian kernel attention computation leads to training difficulties. We first hypothesized that this might be due to lacking normalization (as normalization often helps with training stability and convergence), and tested a variant with softmax on top of an exact Gaussian kernel attention matrix. However, it turns out to suffer from a similar failure. In the time limit we cannot find a solid hypothesis and will keep investigating this problem. Note that in the submission experimental results were included to verify how SOFT works better (see Table 4, main paper), including results obtained by varying the sampling ratio and the sampling strategy.
>
> **Q2: Random sampling and average pooling.**
>
> Thanks for this comment. We have now conducted more experiments to compare the two sampling strategies. The results suggest that random sampling causes training failure when the model size becomes large, whilst average pooling does not have this problem. We will add these new results in the revision.
>
> **Q3: Gaussian weighted self-attention and kernel density estimation.**
>
> Very interesting perspective. Indeed, given that Nystrom is a method to approximate a kernel function, it is reasonable to regard the proposed self-attention as a variant of kernel density estimation. We will add more discussion in the revision.

---

> > ### Comment · Reviewer_aC3r · 2021-08-31
> > **Thank you**
> >
> > Thank you for the detailed comment.
> > I will keep the initial positive score.

---

### Official Review · Reviewer_RD3k · 2021-07-16

**Rating:** 7
**Confidence:** 3

**Summary:**

The paper proposes a softmax-free self-attention that replaces the previous dot-product between the query matrix and the key matrix with a Gaussian kernel. To facilitate the proposed softmax-free self-attention, this paper further proposes to decompose the self-attention matrix with Singular Value Decomposition. The robustness of the proposed low-rank decomposition can be theoretically proved by computing the Moore-Penrose inverse with the Newton-Raphson method.

**Limitations And Societal Impact:**

Yes

**Main Review:**

Introducing kernels/Gaussian kernels to handle self-attention is not new [1]. However, this paper theoretically proves that it’s possible to use low-rank matrix decomposition to approximate such kernel function, which is quite novel and improves the self-attention mechanism. Since I am not an expert in Newton-Raphson methods, I have to kindly ask the other reviewers to verify the correctness of the equations. As for the other parts, the proposed pyramid architecture has outperformed the competitors by large margins in ImageNet classification task and the attention visualizations seem quite intuitive and reasonable to me.

The only question here is on the effectiveness of stacking overlapping convolutions and batch normalizations in L178-L179. It would be better to show some ablation studies comparing the tokenization methods using overlapping/non-overlapping convolution.

[1] Peng, H., Pappas, N., Yogatama, D., Schwartz, R., Smith, N. A., & Kong, L. (2021). Random feature attention. ICLR 2021


**Time Spent Reviewing:**

3 hours

---

> ### Author Response · Authors · 2021-08-10
> **Response to Reviewer RD3k**
>
> We thank the reviewer for the positive and detailed review as well as the suggestions for improvement. Our response to the reviewer’s comments is below:
>
> **Q1: Ablation on overlapping convolution.**
>
> Many thanks. As suggested, we have now ablated SOFT overlapped convolution (our default choice, same as many recent works) and SOFT non-overlapped convolution in our small configuration.
> The results are: 79.3 (overlapped) vs. 77.4 (non-overlapped). Our non-overlapped conv variant still outperforms the PVT model which also has the same non-overlapped conv by a clear margin.
> We will add this in the revised version.
>
> | Methods| overlapped?| Top-1
>   |-----------------|:---------------------:|:----------------------:|
> PVT    |      $\times$       |75.1
> SOFT 	|      $\times$           |77.4
> SOFT | 	✔	|79.3

---

### Official Review · Reviewer_M8ep · 2021-07-16

**Rating:** 7
**Confidence:** 4

**Summary:**

This paper proposes a novel method named SOFT to calculate the token similarity for self-attention without softmax operation. The authors state that the computational complexity could be reduced to O(n) with SOFT. A family of backbones are designed and evaluated on ImageNet, which achieve SOTA performance.

**Limitations And Societal Impact:**

My concerns on the limitations of this paper have been presented in "Main Review". The authors do not discuss the potential negative societal impact of their work. Adding more related discussions will be helpful.

**Main Review:**

This paper follows the work "Nyströmformer: A Nyström-Based Algorithm for Approximating Self-Attention" and modifies its computation of attention score to fit in vision task. But this paper is of novelty.

I have some concerns.

- In 3.2 the authors state that the time complexity is O(m^2n + m^3) and the space complexity is O((2m+d_e)m + m^2). However, as m = n/k, which is linear w.r.t. n, the complexity could not be linear. The complexity could be reduced by k^2 or k according to the order of m. The paper also states that they could keep m as a fixed constant, but with image size increased, based on the method of average pooling or stride convolution, m would increase as well.

- There are some writing problems. The complexity in line 167 lacks m as a multiplier. In table 1 there is no introduction to C33 and C31.

- The convolution layers between stages are not included in many other transformers, and there is no ablation on these convolutional layers.

- There are other works on ViT like botnet, swin, cvt e.t.c. and CNNs like efficientnet that could be add in table 3 for comparison.

- As the complexity decreases with SOFT, there could be some experiments to show that the training and inference speed may increases.

**Time Spent Reviewing:**

8 hours

---

> ### Author Response · Authors · 2021-08-10
> **Response to Reviewer M8ep**
>
> We thank the reviewer for the positive and detailed review as well as the suggestions for improvement. Our response to the reviewer’s comments is below:
>
> **Q1: Bottleneck token length `m`.**
>
> Sorry for the confusion. In this work, `m` is indeed a constant making the complexity linear. This is because in vision tasks, `k` is the downsampling rate, which is typically a variable. In order to make `m` a constant in our experiments for larger images (i.e., larger `n`), we simply increased `k` at the same rate when `n` was increased. As a result, `m` is independent of `n` (we set `m`=49 for all the blocks/stages) and our model’s complexity is thus linear w.r.t. the image/token size as shown clearly in Figure 1(b). We will clarify in the final version.
>
> **Q2: Lack of `m`; C33 / C31.**
>
> Thanks for pointing this out. The missing `m` error will be fixed in the revision.
> The details of C33 (three 3x3 Conv→BN→ReLU, with the stride of 2, 1, 2 respectively) and C31 (one 3x3 conv, down-sampling operation with stride of 2) in Table 1 are given in L179-181. We will make it clearer in the caption.
>
>
> **Q3: Ablation on convolution.**
>
> Unlike the earlier non-pyramidal vision Transformers (e.g., ViT), most recent pyramidal (multi-scale) Transformers  (e.g., PVT) use convolution layers to reduce the spatial resolution (i.e., sequence length) between stages.
> As suggested we have now ablated SOFT with convolution (our default SOFT-Small), SOFT without convolution and ViT (no convolution either).
> Note all three variants have similar parameters and FLOPs.
> Without convolution, we form the model as a non-pyramidal architecture (1/16 resolution), same as the ViT.
> The table below shows that conv-based pyramidal architecture is clearly superior to non-pyramidal design, and our non-pyramidal counterpart is even slightly better than ViT whilst enjoying linear complexity.
> We will add this experiment in the revised version.
>
>
>  |  Methods      |   with Conv  |  Top-1
>   |-----------------|:---------------------:|:----------------------:|
>   |ViT      (non-pyramidal)         |        $\times$             |          79.8           |
>   |SOFT   (non-pyramidal)        |           $\times$                |          80.1           |
>   |SOFT   (pyramidal)        |     ✔                   |         82.2            |
>
>
> **Q4: Bonet, swin, cvt, efficientnet could be add in table 3 for comparisons.**
>
> Thanks. We will include these.
>
> **Q5:  Training and inference speed.**
>
> Good suggestion. As suggested we have now evaluated the throughput (images/sec) of our tiny model for training and inference on a machine with 8 Tesla V100 GPUs. The results are:
>
> |Methods		|Train Throughput (img/s)  |	Inference Throughput (img/s)
> |----------|:--------------:|:----------------:|
> |Transformer		|1073		|		3240
> |Linformer	|	2767		|		3779
> |Performer	|	2037		|		3657
> |Nystromformer	|1891		|		3518
> |SOFT (ours)		|1730		|		3436
>
> From the results, it can be seen that: (1) Our inference speed is on-par with other compared linear Transformers and (2) our training speed is slightly slower than Nystromformer and both are slower than Performer and Linformer. Note that the slow training speed of our model is mostly due to the fact that no official optimized implementation of the Gaussian kernel has yet been made available in Pytorch. It thus can be overcome in the future. So in summary, due to the on-par inference speed we consider the training cost increase is a price worth paying for our superior accuracy.

---

> > ### Comment · Reviewer_M8ep · 2021-09-12
> > **Thank you for the response**
> >
> > I thank the authors for their detailed responses and the additional experiments.
> >
> > The rebuttal address some of my concerns and I will increase my score.

---

### Decision · Program_Chairs · 2021-09-27

**Decision:**

Accept (Spotlight)

**Comment:**

The reviewers unanimously agree that the presented softmax-free transformer and the kernel approximation techniques are novel and interesting. The paper also demonstrates good performance with reduced computation complexity on ImageNet. After the rebuttal, most of concerns from the reviewers are addressed. The authors are encouraged to further improve the paper by 1) comparing with other ViT works/architectures (https://arxiv.org/abs/2103.14030, https://arxiv.org/abs/2101.11605) 2) having more comprehensive discussion regarding related works with softmax-free attention (https://arxiv.org/abs/2103.02143, https://arxiv.org/abs/2103.13076, etc..) 3) discussing potential negative societal impacts of the work.